# Genetic Evaluation in a Cohort of 126 Dutch Pulmonary Arterial Hypertension Patients

**DOI:** 10.3390/genes11101191

**Published:** 2020-10-13

**Authors:** Lieke M. van den Heuvel, Samara M. A. Jansen, Suzanne I. M. Alsters, Marco C. Post, Jasper J. van der Smagt, Frances S. Handoko-De Man, J. Peter van Tintelen, Hans Gille, Imke Christiaans, Anton Vonk Noordegraaf, HarmJan Bogaard, Arjan C. Houweling

**Affiliations:** 1Department of Clinical Genetics, Amsterdam UMC (location VUmc), 1081HV Amsterdam, The Netherlands; l.m.vandenheuvel@amsterdamumc.nl (L.M.v.d.H.); s.alsters@amsterdamumc.nl (S.I.M.A.); J.P.vanTintelen-3@umcutrecht.nl (J.P.v.T.); jjp.gille@amsterdamumc.nl (H.G.); 2Netherlands Heart Institute, 3511EP Utrecht, The Netherlands; 3Department of Genetics, University Medical Centre Utrecht, Utrecht University, 3584CX Utrecht, The Netherlands; j.j.vandersmagt@umcutrecht.nl; 4Department of Lung Disease, Amsterdam UMC (location VUmc), 1081HV Amsterdam, The Netherlands; s.jansen1@amsterdamumc.nl (S.M.A.J.); fs.deman@amsterdamumc.nl (F.S.H.-D.M.); a.vonk@amsterdamumc.nl (A.V.N.); hj.bogaard@amsterdamumc.nl (H.B.); 5Department of Cardiology, St. Antonius hospital, 3435CM Nieuwegein, The Netherlands; m.post@antoniusziekenhuis.nl; 6Department of Cardiology, University Medical Centre Utrecht, Utrecht University, 3584CX Utrecht, The Netherlands; 7Department of Clinical Genetics, University Medical Centre Groningen, 9713GZ Groningen, The Netherlands; i.christiaans@umcg.nl

**Keywords:** pulmonary arterial hypertension, genetic analysis, NGS gene panel, BMPR2, TBX4, GDF2, EIF2AK4

## Abstract

Pulmonary arterial hypertension (PAH) is a severe, life-threatening disease, and in some cases is caused by genetic defects. This study sought to assess the diagnostic yield of genetic testing in a Dutch cohort of 126 PAH patients. Historically, genetic testing in the Netherlands consisted of the analysis of BMPR2 and SMAD9. These genes were analyzed in 70 of the 126 patients. A (likely) pathogenic (LP/P) variant was detected in 22 (31%) of them. After the identification of additional PAH associated genes, a next generation sequencing (NGS) panel consisting of 19 genes was developed in 2018. Additional genetic testing was offered to the 48 *BMPR2* and *SMAD9* negative patients, out of which 28 opted for NGS analysis. In addition, this gene panel was analyzed in 56 newly identified idiopathic (IPAH) or pulmonary veno occlusive disease (PVOD) patients. In these 84 patients, NGS panel testing revealed LP/P variants in BMPR2 (*N* = 4), GDF2 (*N* = 2), EIF2AK4 (*N* = 1), and TBX4 (*N* = 3). Furthermore, 134 relatives of 32 probands with a LP/P variant were tested, yielding 41 carriers. NGS panel screening offered to IPAH/PVOD patients led to the identification of LP/P variants in GDF2, EIF2AK4, and TBX4 in six additional patients. The identification of LP/P variants in patients allows for screening of at-risk relatives, enabling the early identification of PAH.

## 1. Introduction

Pulmonary arterial hypertension (PAH) is a rare and life-threatening disease with an estimated prevalence of 5.9–60 cases per a million adults [1,2,3]. Due to variable and non-specific symptoms, patients are often diagnosed well after the development of right heart failure, indicating a considerable delay in diagnosis.

PAH can be classified into different subgroups, including idiopathic PAH (IPAH), heritable PAH (HPAH), pulmonary veno-occlusive disease (PVOD), drug and toxin induced PAH, and PAH associated with other conditions, including connective tissue disease, human immunodeficiency virus (HIV) infection, portal hypertension, or congenital heart disease [4]. IPAH and PVOD have similar clinical and hemodynamic characteristics but different pathophysiological and histological characteristics and progression rates [5]. IPAH is diagnosed when any potential cause or risk factor has been excluded [4]. HPAH can be diagnosed after the identification of a (likely) pathogenic (LP/P) genetic variant in a PAH associated gene [5]. HPAH generally follows an autosomal dominant inheritance pattern with incomplete penetrance [6]. Both IPAH and HPAH can occur sporadically or familial, when two or more members of a family are affected with PAH.

Pathogenic variants in the *BMPR2* gene (MIM# 131195) were identified as the main genetic cause of HPAH, explaining 75–90% of familial cases and 10–20% of sporadic cases [7,8,9,10]. The *BMPR2* gene belongs to the transforming growth factor β (TGF-β) superfamily and is involved in the regulation of cell growth and apoptosis. Evans et al. showed that compared to *BMPR2*-negative PAH patients, carriers of a pathogenic variant in *BMPR2* were, on average, diagnosed at a younger age and had a more severe progression of the disease [7].

Until 2018, DNA diagnostics in IPAH patients in the Netherlands consisted of sequencing *BMPR2* and *SMAD9* (OMIM # 615342), encoding an intracellular signal tran Material S2ucer of the TGF-B pathway. The additional use of next-generation sequencing (NGS) techniques has resulted in the identification of several additional genes potentially associated with PAH [9]. However, these recently identified gene–disease relationships remain to be established beyond case-control studies and with more certainty over time. In a minority of IPAH cases, (potentially) associated variants have been reported in several genes including *KCNK3* (encoding a PH sensitive potassium channel), *CAV1* (encoding an integral membrane protein), *TBX4* (playing a critical role in the development of respiratory system), *GDF2* (also called BMP9 and a member of the highly conserved transforming growth factor-β superfamily), *AQP1* (encoding aquaporin-1), *SOX17* (encoding a transcription factor involved in Wnt/β-catenin and Notch signaling during development), and *BMPR1B* (encoding a type I BMP receptor) [5,9,11]. *EIF2AK4* variants have been reported as an autosomal recessive cause of PVOD whereas variants in the *BMPR2* gene have also been reported to cause PVOD [12,13,14].

The identification of a pathogenic variant in a patient diagnosed with IPAH/PVOD allows for the early detection of disease in genotype positive relatives. This enables timely treatment, which is important as a recent study indicated that early treatment of PAH delays disease progression and improves transplant-free survival [15]. Furthermore, predictive DNA testing enables reproductive choices, including the possibility of preimplantation genetic diagnosis [8].

Previous research has shown that genetic testing may reveal a genetic predisposition to the disease also in sporadic cases. In two studies, disease-causing variants were identified in 13–17% of sporadic IPAH/PVOD cases [8,16]. This can likely be explained by the relatively low penetrance in HPAH and the de novo occurrence of pathogenic variants [17,18,19]. While newly diagnosed IPAH patients can benefit from genetic counselling and DNA testing, only a limited number of patients undergo genetic testing and this is generally limited to *BMPR2/SMAD9*. The recent identification of additional genes involved in the disease, in combination with increasing evidence on health benefits of early detection of the disease in unaffected relatives, prompted us to offer all IPAH/PVOD patients genetic counselling and (extended) genetic testing using an NGS panel of 19 PAH-associated genes if necessary. Here, we describe the diagnostic yield of genetic testing in a Dutch cohort, in terms of identified (likely) pathogenic variants and the number of identified genotype positive relatives.

## 2. Materials and Methods

### 2.1. Subjects

Genetic testing has been offered to IPAH/PVOD patients referred to the PAH center of expertise in the Amsterdam UMC since 2002. Prior to the introduction of NGS technologies, genetic testing was not routinely offered to all patients but often only in familial cases. Patients with IPAH or PVOD were included in this study from January 2018 until April 2020. Patients in whom prior analysis of BMRP2 and SMAD9 did not result in the detection of a LP/P variant and patients who were not previously tested were offered the option of (extended) genetic testing using NGS via a letter of their treating physician. Interested patients received genetic counselling and could opt for genetic testing with an NGS panel consisting of 19 genes. All included patients were unrelated. The clinical geneticist or genetic counsellor obtained informed consent for genetic testing from all patients. All patients provided consent for the use of their data for research. The study was conducted in accordance with the Declaration of Helsinki. The Medical Ethical Committee of the Amsterdam UMC (location VUmc) assessed the study protocol and confirmed that the study was exempt from ethics review according to the Dutch Medical Research Involving Human Subjects Act (2017.541).

All patients in whom a LP/P variant as a cause of their PAH was detected were encouraged to inform their first-degree relatives about the option of predictive DNA testing. For assistance, we provided personalized family letters for patients’ relatives with information about the disease and access to genetic counselling and—if desired—predictive DNA testing. Patients in whom a causal BMPR2 mutation had been detected via genetic testing (i.e., HPAH patients) prior to January 2018 were informed about current preventive and treatment options for themselves and their at-risk relatives by letter, including the information on preimplantation genetic diagnosis (PGD). They were also asked to inform their relatives, supported by an updated family letter if desired. Relatives could opt for genetic testing when interested.

### 2.2. Data Collection

Sociodemographic data (age at diagnosis and sex), clinical data (NYHA functional class, mean pulmonary artery pressure (mPAP), mean right atrial pressure (mRAP), pulmonary vascular resistance (PVR) and right ventricle end-diastolic volume index (RVEDVI), right ventricle end-systolic volume index (RVESVI), and right ventricular ejection fraction (RVEF) at diagnosis), as well as family history were collected. DNA diagnostics was performed at the DNA laboratory of Amsterdam UMC, location VUmc. Up until 2017, DNA testing of BMPR2 and SMAD9 were performed using Sanger sequencing. From 2018 onwards, a WES-based virtual panel was analyzed, which included 19 PAH-associated genes (*ABCA3, ACVRL1, BMPR1B, BMPR2, CAV1, EIF2AK4, ENG, FOXF1, GDF2, KCNA5, KCNK3, NOTCH1, NOTCH3, RASA1, SMAD1, SMAD4, SMAD9, TBX4*, and *TOPBP1*). AQP1 and SOX17 were not included in this NGS panel. Using the NGS test point, mutations and small insertions and deletions can be detected. Additionally, multiplex ligation-dependent probe amplification (MLPA) was used to detect large deletions or duplications of BMPR2, because our WES-based panel test does not allow the detection of exon deletions and duplications.

All variants detected were classified using the ACGS/VKGL guidelines [20]. For variant classification, a 5-class variant classification system was used: class 1 (benign), class 2 (likely benign), class 3 (variant of uncertain significance), class 4 (likely pathogenic), and class 5 (pathogenic) [21]. The classification of variants was based on the occurrence of the variant in control populations (gnomAD database), in silico predictions of the impact of an amino acid change on the function of the protein (PolyPhen2, SIFT, AlignGVGD), and in silico prediction of the potential impact of the nucleotide change on splicing. Variants causing frameshifts or premature stop codons were considered likely pathogenic (class 4) or pathogenic (class 5). Variants occurring in a control population at a frequency > 1% were considered polymorphisms (class 1). Detailed information on the analysis methods is given in Appendix A.

### 2.3. Statistical Analysis

Statistical analysis was performed with SPSS (Version 25.0) and R Studio (Version 4.0.2, 2020-06-22). Data were visualized using the R ggplot2 package. Data were described as mean and SD, or median and IQR, as appropriate. Chi-square tests, log-rank tests, t-tests, and Wilcoxon signed rank tests were used to assess the differences between clinical observations in patients with and without a (likely) pathogenic variant, and between patients with a positive and negative family history, where appropriate. A *p*-value of <0.05 was considered statistically significant.

## 3. Results

### 3.1. Genetic Analyses

Figure 1 shows a flow-chart of the number of patients in whom genetic testing was performed. In total, 126 patients were included in our study. Until 2018, genetic testing at the Amsterdam UMC consisted of analysis of BMPR2 and SMAD9. These genes were analyzed in 70 of the 126 patients. In 22 of these 70 patients (31%), a LP/P variant of BMPR2 was identified. Additional genetic testing (NGS panel) was offered to the 48 BMPR2 and SMAD9 negative patients, including 2 patients with a VUS in the BMPR2 gene. Of them, 28 opted for NGS analysis. Fifty-six patients diagnosed with PAH after 2018 were directly tested with this NGS panel. In total, 84 patients were genetically tested using NGS. The NGS panel yielded 10 additional patients with a LP/P variant, 3 LP/P variants in the PAH patients previously tested negative for LP/P variants in BMPR2 and SMAD9 and 7 LP/P variants in the PAH group diagnosed after 2018. NGS panel testing revealed LP/P variants in BMPR2 (*N* = 4), GDF2 (*N* = 2), EIF2AK4 (*N* = 1), and TBX4 (*N* = 3). In 11 out of 84 patients (9%), NGS panel revealed a VUS (class 3 variants). In total, a LP/P variant was identified in 32 out of 126 patients (25%).

Table 1 shows the characteristics of the study population. The mean age at diagnosis was 49 years (*SD* = 16) in all patients, with a mean age of 52 years in patients without and 37 years in patients with a LP/P variant (*p* = 0.001). Seventy-one percent of patients were female. The majority of patients (*N* = 114, 90%) were diagnosed with IPAH, whereas only 12 patients were diagnosed with PVOD. Seventeen patients in our total cohort had a positive family history for PAH. Of the patients with a LP/P variant (*n* = 32), nine had a positive family history. In eight patients with a positive family history no LP/P variant could be detected. A negative family history was reported in 13 patients (41%) with a LP/P variant compared to 55 patients (59%) without a LP/P variant. During follow-up (median 3 years, range 0–7), seventeen patients died, and four patients received a lung transplant.

The variants in the group of patients with a LP/P (*N* = 32) were identified in BMPR2 (*N* = 26), TBX4 (*N* = 3), GDF2 (*N* = 2), and EIF2AK4 (*N* = 1), see Table 2. One identical pathogenic variant in BMPR2 was identified in three unrelated probands (c.1471C > T, p.(Arg491Trp)), indicating a potential founder effect of this variant. VUS were found in FOXF1 (*N* = 3), NOTCH3 (*N* = 4), BMPR2 (*N* = 3), and TBX4 (*N* = 2), of whom two VUS were found in one patient (see Appendix A).

### 3.2. Genotype-Phenotype Correlation

The main clinical characteristics of the patients with and without a LP/P variant identified are presented in Table 1. A significant difference in age at diagnosis in patients with and without a LP/P variant was observed, those with a LP/P variant had a significantly younger age at diagnosis (*p* = 0.001), as shown in Figure 2. In addition, patients with a LP/P variant had significantly higher mPAP and PVR at diagnosis compared to those without LP/P variant (*p* = 0.008 and *p* = 0.022, respectively). Moreover, a significantly higher RVESVI and significantly lower RVEF was observed in patients with a LP/P variant, (*p* = 0.038 and *p* < 0.001 respectively). Furthermore, a LP/P variant was identified more often in patients with a positive family history compared to those with a negative family history (*p* = 0.025).

### 3.3. Relatives

Following the initial identification of LP/P variants in 32 probands, 134 relatives (range per family: 0–42) have been tested to date. Of the relatives pursuing predictive DNA testing, 41 were shown to be a carrier, of whom 36 relatives carried a variant in BMPR2 (88%), 3 in TBX4 (7%), and 2 in GDF2 (5%). These relatives were offered annual check-ups at the out-patient clinic to detect early signs of PAH and receive treatment accordingly. In one asymptomatic carrier, mild PAH was diagnosed at the first visit and treatment was subsequently started. Further cascade genetic testing and clinical screening is currently ongoing. One relative, who had opted for predictive DNA testing and turned out to be a carrier, successfully pursued PGD.

## 4. Discussion

In this study, we describe the results of genetic testing and the characteristics in a Dutch cohort of 126 adult probands diagnosed with non-associated PAH or PVOD. A LP/P variant was identified in 25% of patients in this cohort. The vast majority of these patients had a LP/P variant in *BMPR2* (81%). LP/P variants were detected in *BMPR2* (*N* = 26), *GDF2* (*N* = 2), *EIF2AK4* (*N* = 1), and *TBX4* (*N* = 3). Expanding genetic testing in 28 patients previously tested negative for LP/P variants in *BMPR2* and *SMAD9* resulted in the identification of three disease-causing variants (11%). Patients with a LP/P variant had worse hemodynamics and a younger age at diagnosis.

In eight families with familial PAH, we were unable to identify a disease-causing variant with our NGS panel, pointing to the possibility of other genetic causes [27]. In these gene-elusive families, first-degree relatives of PAH patients were offered clinical screening. Furthermore, these patients and other PAH patients were asked to participate in international efforts aimed at the identification of novel PAH associated genes [9,28,29]. With these international efforts, an *AQP1* variant (c.583C > T; p.Arg195Trp) was identified in one PAH patient who tested negative on our NGS panel [9]. Novel gene–disease relations are established at a rapid pace for PAH and other genetic diseases. Especially when the identification of a genetic cause can result in health benefits in relatives, it is important to establish patient databases and to obtain informed consent by which mutation negative patients can be re-contacted for future additional genetic testing.

In our cohort, three patients had a disease-causing variant in *TBX4*. Variants in *TBX4* have previously been recognized as a cause of neonatal and paediatric pulmonary hypertension [30,31]. However, recent studies also reported pathogenic *TBX4* variants in adult-onset pulmonary hypertension [9,16]. We previously described the clinical characteristics of our *TBX4* patients indicating a female predominance, bronchial diverticulosis, distinct skeletal anomalies, and a history of asthma in all [32]. PAH associated with variants in *TBX4* is clinically highly variable [33]. In addition to *TBX4*, we identified LP/P variants in *GDF2*, resulting in loss of *BMP9* function [34]. Causal variants in *GDF2* (*BMP9*) were first described in adult-onset PAH patients by Gräf et al. [9], and subsequently reported in two studies describing *GDF2* variants in respectively 6.7% and 1.1% of sporadic PAH patients [31]. Interestingly, no LP/P variants were found in the other 15 PAH-associated genes, including *SMAD9*. Although *SMAD9* has been repeatedly shown to cause PAH when disrupted, it is considered to be a very rare cause of PAH. This is confirmed by the absence of LP/P variants in this gene in our cohort.

In this study, only *BMPR2* was screened for larger deletions or duplications using MLPA. Therefore, larger deletions or duplications in the other genes on our panel may have been missed. Larger Copy Number Variants (CNVs) including the *TBX4* gene have been reported to cause PAH [4]. As these CNVs often include additional genes located in proximity to the TBX4 gene, they often cause additional features, such as intellectual disability and congenital defects. It is therefore unlikely that large CNVs are common in the cohort reported here, as such features were not present. Small deletions/CNVs, however, may have been missed in our cohort. In a previous study in patients with hereditary thoracic aortic aneurysms small CNVs were identified as a genetic cause in 6/66 (9%) patients with a LP/P variant using the exome hidden Markov model (XHMM; an algorithm to identify CNVs in targeted NGS data) [35]. Subsequent cohort studies are required to further establish gene–disease relations with certainty and to elucidate the role of small CNVs in PAH genes.

In conclusion, genetic testing in a Dutch cohort of 126 non-associated PAH/PVOD patients revealed a LP/P variant in 32 patients (25%). BMPR2 was the main cause (88%) of the LP/P variants. NGS identified a genetic cause in an additional six patients. *TBX4* and *GDF2* variants were found in three and two patients with PAH, respectively. In addition, a homozygous variant in *EIF2AK4* was identified in one PVOD patient. A genetic cause was identified in 21% of sporadic cases, underscoring the importance of genetic testing in PAH/PVOD. The identification of LP/P variants in patients allows for screening of at-risk relatives; in this study, 41 out of 134 unaffected tested relatives (31%) were shown to be a carrier. Predictive DNA testing allows for clinical screening of at-risk relatives, supporting the early identification of PAH and the possibility of PGD.

## Figures and Tables

**Figure 1 genes-11-01191-f001:**
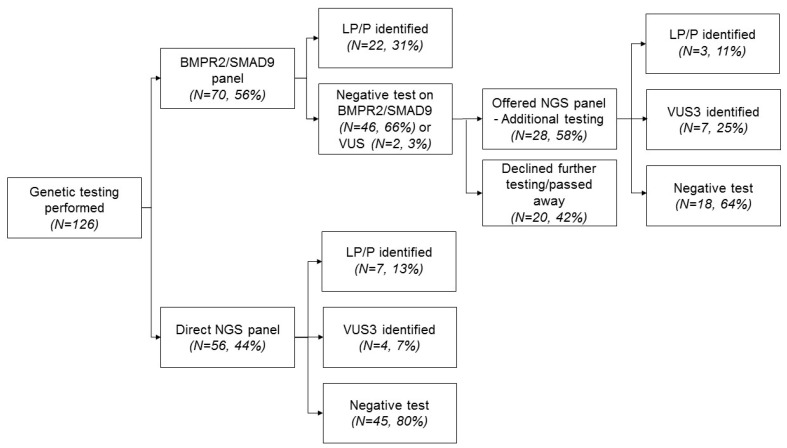
Flow-chart of genetic analyses in idiopathic/pulmonary veno occlusive disease (IPAH/PVOD) patients.

**Figure 2 genes-11-01191-f002:**
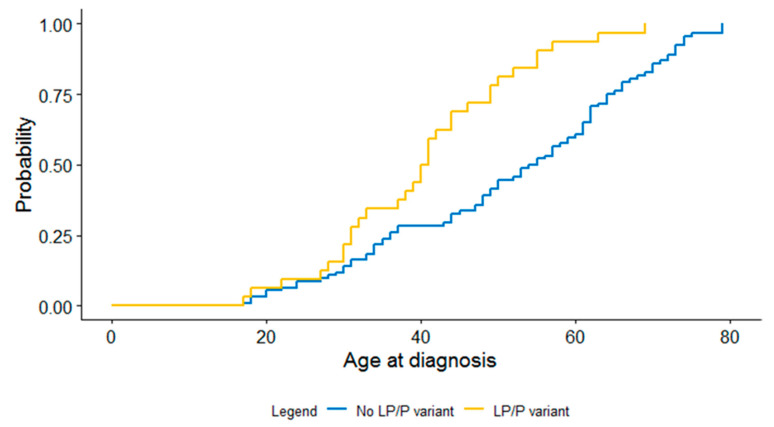
Kaplan–Meier curve for difference in age at diagnosis between patients with and without a LP/P variant.

**Table 1 genes-11-01191-t001:** Sociodemographic and clinical characteristics of patients eligible for genetic testing at diagnosis.

	All Patients *N* = 126	No LP/P Variant *N* = 94	LP/P Variant *N* = 32	
Characteristic	*N* (%)	*N* (%) ^a^	*N* (%) ^a^	*p* Value
Sex				
Female	89 (71)	66 (70)	23 (72)	0.922
Male	37 (29)	28 (30)	9 (28)	
Age at diagnosis, years	49 ± 16	52 ± 17	37 ± 13	0.001
Clinical diagnosis ^b^				
IPAH	114 (90)	84 (89)	30 (94)	0.031
PVOD	12 (10)	10 (11)	2 (6)	
NYHA functional class				
NYHA I-II	36 (29)	25 (29)	11 (34)	0.587
NYHA III-IV	78 (62)	60 (71)	18 (56)	
BMI, kg/m^2^	26.9 ± 5.2	27.0 ± 5.3	26.6 ± 5.0	0.757
Hemodynamics				
mPAP, mmHg	52 ± 17	50 ± 17	59 ± 14	0.008
PVR, WU	9.9 (5.7–12.7)	8.8 (4.8–12.2)	11.9 (9.2–15.2)	0.022
mRAP, mmHg	9 (6–11)	9 (6–11)	8 (6–11)	0.214
PCWP, mmHg	10 ± 3	10 ± 3	9 ± 3	0.039
CI, L/min/m^2^	2.5 ± 0.8	2.6 ± 0.8	2.2 ± 0.6	0.061
RVEDVI, mL/m^2^	84 ± 27	82 ± 27	90 ± 27	0.243
RVESVI, mL/m^2^	53 ± 26	50 ± 25	64 ± 28	0.038
RV EF, %	38 ± 13	41 ± 12	30 ± 12	<0.001
Family history				
No	68 (54)	55 (59)	13 (41)	0.025
Yes	17 (13)	8 (9)	9 (28)	
Unclear	31 (25)	22 (23)	9 (28)	
Unknown	10 (8)	9 (10)	1 (3)	
Death	17 (14)	11 (12)	6 (20)	0.324
Lung transplant	4 (3)	4 (4)	0 (0)	NA ^c^
Median FU, years	3 (0–7)	2 (0–5)	6 (2–11)	0.001

Data are given as mean (SD), median (range) or number (percentage). IPAH = Idiopathic pulmonary arterial hypertension; PVOD = pulmonary veno-occlusive disease; NYHA = New York Association functional class; BMI = body mass index; mPAP = mean pulmonary arterial pressure; PVR = pulmonary vascular resistance; mRAP = mean right atrial pressure; RVEF = right ventricular ejection fraction; RDEVI = right ventricular end-diastolic volume index; RVESVI = right ventricular end-systolic volume index. ^a^ Not all numbers add up to the total number of patients due to missing values. ^b^ Concerns *p* value of chi-square test performed on difference ‘idiopathic PAH’ versus ‘PVOD’, due to >20% of cells having an expected count less than 5. ^c^ Significance testing not possible due to small numbers.

**Table 2 genes-11-01191-t002:** Overview of the detected class 4 and 5 variants in 32 unrelated pulmonary arterial hypertension (PAH) patients.

Gene	Nucleotide Change	Amino Acid Change	Class	Novel	Reference	Remark Pathogenicity
*TBX4 **	c.40_49del	p. (Phe14Argfs*28)	Class 5 ^a^	No	[9]	NA
*TBX4 **	c.916G > T	p.(Glu306 *)	Class 4	Yes	NA	Premature stopcodon; not present in gnomAD and ClinVar
*TBX4 **	c.1112del	p.(Pro371Leufs*8)	Class 5 ^a^	No	[10]	NA
*BMPR2*	c.(?_-1)_(*1_?)del (entire gene)	p.0	Class 5	No	[22]	NA
*BMPR2*	c.(?_-1)_(76 + 1_77-1)del (exon 1)	p.? ^e^	Class 5	No	[13] ^a^	
*BMPR2*	c.76 + 2T > G	p.? ^e^	Class 5	No	[9] ^a^	NA
*BMPR2*	c.246A > G	p.(Glu48_Gly83del) (splice defect)	Class 4	Yes	NA	Defective splice donor site exon 2; use of a cryptic splice donor site in exon 2 (mRNA analysis performed in our lab)
*BMPR2*	c.348C > G	p.Cys116Trp	Class 4	Yes	NA	Variant not present in controls (gnomAD). Highly conserved region; AlignGVGD: class C65; SIFT: deletious; PolyPhen2: probably damaging, score 1.000. Variant not present in ClinVar
*BMPR2*	c.350G > C	p.(Cys117Ser)	Class 5	No	[13]	NA
*BMPR2*	c.399del	p.(Pro134Leufs*18)	Class 5	No	[23] ^a^	NA
*BMPR2*	c.619dup	p.(Glu207Glyfs* 13)	Class 5	No	[23] ^a^	NA
*BMPR2*	c.690del	p.(Val231Cysfs*21)	Class 5	Yes	[23] ^a^	NA
*BMPR2*	c.852_852 + 1insA	p.(Gly285Argfs*13)	Class 4	No	[9]	NA
*BMPR2*	c.994C > T	p.(Arg332 *)	Class 5	No	[24]	NA
*BMPR2*	c.1133G > T	p.(Gly378Val)	Class 5 ^c^	No	[9]	NA
*BMPR2*	c.1217T > G	p.(Met406Arg)	Class 4	No	[9]	NA
*BMPR2*	c.1454A > G	p.(Asp485Gly)	Class 5	No	[23] ^a^	NA
*BMPR2*	c.1459G > T	p.(Asp487Tyr)	Class 4 ^c^	Yes	NA	Not present in controls (gnomAD); Highly conserved region; AlignGVGD: class C65; SIFT: deletious; PolyPhen2: probably damaging, score 1.000. ClinVar: 1 entry, likely pathogenic (VCV000212812.2)
*BMPR2* *(N = 3)*	c.1471C > T	p.(Arg491Trp)	Class 5	No	[23] ^a^	NA
*BMPR2*	c.1487G > A	p.(Cys496Tyr)	Class 5	No	[13]	NA
*BMPR2*	c.1525G > T	p.(Glu509 *)	Class 5	No	[25] ^a^	
*BMPR2*	c.1978G > T	p.(Glu660 *)	Class 5	No	[25] ^a^	
*BMPR2*	c.2161C > T	p.(Gln721 *)	Class 5	Yes	NA	Premature stopcodon. Not present in gnomAD or ClinVar
*BMPR2*	c.2752C > T	p.(Gln918 *)	Class 5	No	[25] ^a^	
*BMPR2*	c.(418 + 1_419-1)_(2866 + 1_2867-1)del (exon 4-12)	p.? ^e^	Class 5	No	[23] ^a^	NA
*BMPR2*	c.(529 + 1_530-1)_(967 + 1_968-1)del (exon 5-7)	p.? ^e^	Class 5	No	[13]	NA
*BMPR2*	c.(967 + 1_968-1)_(1128 + 1_1129-1)dup (exon 8)	p.(Val377Ilefs*12)	Class 5	No	[23] ^a^	A tandem exon duplication was confirmed by Sanger sequencing on cDNA
*EIF2AK4*	c.1739dup	p.(Arg581Glufs*9)	Class 5 ^d^	No	[26]	NA
*GDF2*	c.328C > T	p.(Arg110Trp)	Class 4	No	[9] ^b^	NA
*GDF2*	c.451C > T	p.(Arg151 *)	Class 4	No	[16]	NA

NA not applicable. Reference sequences: BMPR2 NM_001204.6; EIF2AK4 NM_001013703.3; GDF2 NM_016204.3 and TBX4 NM_018488.2. ^a^ These patients only tested on BMPR2 and SMAD9 were reported previously in Van der Bruggen et al. [23] and/or Girerd et al. [25]. ^b^ This patient was previously reported by Gräf et al. [9].^c^ Both parents tested negative for this variant. ^d^ Homozygotic. ^e^ An effect on the protein level is expected, but it is not possible to give a reliable prediction of the consequences. * Was previously tested on BMPR2/SMAD9.

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
