# Peer review of "Genetic Evaluation in a Cohort of 126 Dutch Pulmonary Arterial Hypertension Patients"

_genes, 2020, doi:10.3390/genes11101191_

Round 1

Reviewer 1 Report

In this manuscript titled, " Next generation gene panel analysis in a cohort of 126 Dutch pulmonary arterial hypertension patients ", Lieke M. van den Heuvel et al., authors assess the diagnostic yield of a next generation sequencing (NGS) panel of PAH-associated genes. This manuscript appears preliminary. 1. Authors should rewrite Abstract. The title describes next generation gene panel analysis in 126 PAH patients, but in abstract only 28 patients were opted for NGS panel testing. The statement from line 26 to 31 is very confused. 2. If a full NGS analysis was performed on these samples, then this data and proper analysis needs to be shown and discussed. 3. The 19 selected PAH-associated genes should be further confirmed by WB or qPCR using the same patients' samples.

Reviewer 2 Report

The authors report the results of a comprehensive screen of 19 PAH-associated risk genes in 84 patients without prior genetic diagnosis. Likely pathogenic or pathogenic variants were identified in 21% of sporadic cases, indicating the utility of genetic screening for these patients. In addition, screening of unaffected relatives identified individuals at risk who were offered annual clinical screening for early PAH detection and informed young adults in terms of family planning decisions.

Specific comments:

Introduction - The 35-40% frequency of BMPR2 variants among sporadic PAH cases seems high. In the referenced papers, the 26% estimate in Evans et al included familial cases and the estimate for sporadic cases in Girerd et al was 12%. Estimates from recent registries, 10-20% including the UK NIHR Bioresource (Graf et al 2018) and the PAH Biobank (Zhu et al 2019), are close to the Girerd estimate.

Methods - Additional description of variant filtering/classification criteria would be helpful. Was a population allele frequency threshold applied? Which deleterious prediction tools were used for missense variants and what threshold scores were used to define pathogenicity classifications? A brief description (1-2 sentences) of  the threshold variables used to distinguish class 5, 4 and 3 variants would be helpful for readers unfamiliar with the ACGS/VKGL guidelines.

Results, Table 2 - Addition of population allele frequency from gnomAD and CADD score (or other deleterious prediction tools) for each variant would increase the transparency of the pathogenicity score. Is the mode of inheritance known from any of the relative analyses? In particular, any variants known to be de novo should be noted. However, since this is a mostly adult-onset cohort, parental samples may not be available and likely the focus was on informing subsequent generations. In regards to previously-reported variants, the authors might also check Machado 2015 and Zhu 2019 for extensive lists of BMPR2 variants. A number of the variants in Table 2 were reported in these papers, including the recurrent BMPR2 variant (c.1471C>T).

Discussion - The Discussion is a bit repetitive with the Introduction and Results and could be slightly shortened.

Round 2

Reviewer 1 Report

The revised manuscript has significantly improved and the data are enough for current study.

Author Response

We thank the reviewer for reviewing the manuscript again and appreciate that he/she is of the opinion that our adjustments have significantly improved the manuscript and that the data are enough for the current study. Minor changes have been made to Table 2, according to the editor's comments.